# Preparation and Characterization of Microalgae Styrene-Butadiene Composites Using *Chlorella vulgaris* and *Arthrospira platensis* Biomass

**DOI:** 10.3390/polym15061357

**Published:** 2023-03-08

**Authors:** Marius Bumbac, Cristina Mihaela Nicolescu, Radu Lucian Olteanu, Stefan Cosmin Gherghinoiu, Costel Bumbac, Olga Tiron, Elena Elisabeta Manea, Cristiana Radulescu, Laura Monica Gorghiu, Sorina Geanina Stanescu, Bogdan Catalin Serban, Octavian Buiu

**Affiliations:** 1Faculty of Science and Arts, Valahia University of Targoviste, 13 Aleea Sinaia, 130004 Targoviste, Romania; 2Institute of Multidisciplinary Research for Science and Technology, Valahia University of Targoviste, 13 Aleea Sinaia, 130004 Targoviste, Romania; 3National Research and Development Institute for Industrial Ecology-ECOIND, 57-73 Drumul Podu Dambovitei, District 6, 060652 Bucharest, Romania; 4IMT Bucharest, National Institute for Research and Development in Microtechnologies, 126A Erou Iancu Nicolae, 077190 Voluntari, Romania

**Keywords:** elastomers, microalgae polymer composites, biogenic materials, microalgae, cyanobacterium, algal biomass

## Abstract

The food industry is a high consumer of polymer packing materials, sealing materials, and engineering components used in production equipment. Biobased polymer composites used in the food industry are obtained by incorporating different biogenic materials into the structure of a base polymer matrix. Renewable resources such as microalgae, bacteria, and plants may be used as biogenic materials for this purpose. Photoautotrophic microalgae are valuable microorganisms that are able to harvest sunlight energy and capture CO_2_ into biomass. They are characterized by their metabolic adaptability to environmental conditions, higher photosynthetic efficiency than terrestrial plants, and natural macromolecules and pigments. The flexibility of microalgae to grow in either low-nutrient or nutrient-rich environments (including wastewater) has led to the attention for their use in various biotechnological applications. Carbohydrates, proteins, and lipids are the main three classes of macromolecular compounds contained in microalgal biomass. The content in each of these components depends on their growth conditions. In general, proteins represent 40–70% of microalgae dry biomass, followed by carbohydrates (10–30%) and lipids (5–20%). A distinctive feature of microalgae cells is the presence of light-harvesting compounds such as photosynthetic pigments carotenoids, chlorophylls, and phycobilins, which are also receiving growing interest for applications in various industrial fields. The study comparatively reports on polymer composites obtained with biomass made of two species of green microalgae: *Chlorella vulgaris* and filamentous, gram-negative cyanobacterium *Arthrospira*. Experiments were conducted to reach an incorporation ratio of the biogenic material into the matrix in the 5–30% range, and the resulting materials were characterized by their mechanical and physicochemical properties.

## 1. Introduction

A significant development of research in the field of food packaging has been achieved over the past few years. Two notable reasons may be mentioned in this respect: on the one hand, they have a negative contribution insofar as they increase the waste volume, which has a negative environmental impact, and on the other hand, interest in extending the shelf life of semiprepared foods has increased, especially in more-developed countries. The market size of manufacturing food packaging materials was evaluated at USD 346.5 billion in 2021, and the previewed growth rate for the period 2022 to 2030 was estimated at 5.5% [1]. The evolution of food preparation habits and consumption, in the context of the positive development of many areas and markets around the world, includes several aspects that correlate with the positive evolution of the food packaging business [2]. Among the plastic consumers, the packaging industry is at the top, with the largest volume worldwide, and it consequently became the main source of plastic waste that accumulate in the surroundings.

The use of biodegradable polymers has arisen as a good answer to this problem, and polymer composite producers have reoriented their industrial capacities in this direction. Although the biobased polymers market has increased in the past few years, the degree of biodegradability is still at low levels. In the past few years, significant research has been performed that has aimed at obtaining composite polymer materials with high biodegradation potential [3,4,5]. On the other hand, the food industry uses polymer materials as sealing materials and other engineering components for production equipment, and these must meet specific physical-mechanical requirements, as well as increased resistance to chemical substances under the conditions of food industrial processes [6].

Elastomers represent a class of polymer materials with different structures, which have characteristics that make them useful in the manufacture of sealing gaskets, gloves, tees, conveyor belts, and tubes used in the food industry. These materials may be in contact with food and food additives, and it is desirable that they can be recycled at the end of their life cycle [7].

Poly (styrene-butadiene-styrene) copolymers (SBS) and styrene-butadiene copolymers (SBC) are classes of hard rubbers used in the manufacture of polymer elements used in materials that are capable of maintaining their tensile shape and are durable/resistant to wear. SBS and SBC are both fossil-based polymers. SBS is a type of block copolymer that consists of three segments. The first is a long polystyrene chain, the middle is a long polybutadiene chain, and the last is another long polystyrene section [8]. Polystyrene is hard plastic, while polybutadiene is rubbery—both components providing durability and elasticity to the SBS copolymer [9]. In addition to these properties of the two polymers, the ability of the polystyrene chains to agglomerate thanks to the π–π interactions between the aromatic nuclei is noticeable as they may produce clumps that can accommodate bioconstituents in the polymer matrix [10]. SBS polymers consist of polystyrene glass domains connected by polybutadiene segments, thus having a two-phase morphology. The groups of styrene chains of the SBS molecule are joined through polybutadiene rubber chains, an interaction resulting their ability to retain their shape after stretching [11,12].

Microplastics (plastic particles smaller than 5 mm) have gained more attention as they have been found to cause irreparable damage in various ecosystems [13]. Additionally, it was observed that microplastics can agglomerate toxic contaminants in water, including heavy metals and organic pollutants [14,15]. Therefore, accumulated plastics and toxic substances can enter various food chains (terrestrial and aquatic) and eventually find their way into the human body through trophic transfer, thus posing several potential risks to human health [16,17]. The industrial use of plastic has also raised many environmental concerns, particularly with regard to the release of microplastics into the atmosphere and water systems. The plastic particles come from industrial raw materials such as pellets, spherules, granules, and other plastic materials. Other harmful plastic waste generated by industrial plants in the environment include solvents, lubricating oils, and washing liquids resulting from industrial equipment [18].

Incorporating biogenic materials into polymer matrices may lead to the physicochemical and mechanical properties that make composites suitable for industrial applications. Additionally, in terms of environmental effects, given the results that have been reported in some studies [19,20,21], one may assume that the higher the content of biogenic materials incorporated in polymer composites, the higher the biodegradability of the final plastic materials.

The microalgae biomass generally contains carbohydrates, lipids, proteins, and pigments and therefore are valuable potential fillers for polymers [22,23]. In addition, the benefit of using microalgae for plastic production is higher than the one offered by the terrestrial crops’ biomass. Thus, it does not compete with food applications, and microalgae biomass has a rapid development rate and relatively low production costs [24,25].

*Chlorella vulgaris* is a green, unicellular, microscopic eukaryotic alga with high nutritional value. The dried biomass of *Chlorella vulgaris* is presented in the form of a powder, containing lipids (5–40% dry weight) [26], carbohydrates (12–55% dry weight) [27], and proteins (20–65% dry weight) [28]; it is used in biotechnology mainly as a food supplement. Compared with *Chlorella* sp., *Arthrospira platensis* is a blue-green, multicellular, filamentous, spiral cyanobacterium [29,30] that is rich in protein (40–70% dry weight) and lower in fat and carbohydrate [31].

An important phase in the preparation of polymer composites consists of the extrusion of the mechanically previously mixed components, a process conducted at relatively high temperatures, near the melting point of the polymer matrix, so that the biomass can be incorporated into the polymer matrix [32]. The structure of the algal biomass can change as an effect of the heat and pressure applied in this extrusion phase.

The present study aims at investigating the physical, chemical, and mechanical particularities of the polymer composites resulting from the incorporation of the biogenic materials of two types of microalgae species, with different morphologies, structures, and chemical compositions (*Chlorella vulgaris* and *Arthrospira platensis*), into SBS polymer matrices.

## 2. Materials and Methods

### 2.1. General

Analytical purity ethanol (96%, evaporation residue ≤25 mg/L), acetone (purity ≥99.8%, evaporation residue ≤3 mg/L), methanol (anhydrous, purity ≥99.8%) provided by the local representative of Merck Millipore were used to prepare the testing solutions, namely 90% (*v*/*v*) solutions of these three organic solvents.

Redistilled water (<0.1 μS·cm^−1^ at 25 °C) was used for all the experiments.

The styrene-butadiene-styrene copolymer used in the preparation of the MSBS (microalgae styrene-butadiene-styrene) composite has a linear structure and was obtained by using the block synthesis of styrene and butadiene in the presence of an alkyllithium catalyst.

Two types of SBS copolymers with different amounts of styrene content were used. One of the constituents had 30% (*w*/*w*) styrene and 70% (*w*/*w*) butadiene content (SBS 1), and the other one had 40% styrene (*w*/*w*) and 60% (*w*/*w*) butadiene (SBS 2). Both copolymers were purchased from LCY Grit Corporation, Taiwan. Paraffin oil (density 0.87 g/cm^3^ and viscosity 18 cSt at 40 °C), which was used as a plasticizer, was purchased from Apar Ltd., Mumbai, India.

*Arthrospira platensis* (formerly *Spirulina platensis*, trade name Spirulina) and *Chlorella vulgaris* (trade name Chlorella) powders were purchased from local suppliers (Rawboost Smart Food SRL, Targoviste, Romania and Hyperici Pharm SRL, Targoviste, Romania, respectively). Both powders had a moisture content of 6.0 ± 0.5% (*w*/*w*), which was low enough to allow their respective incorporation into the composite polymer matrices.

### 2.2. Preparation of Polymer Composites

The study aimed at evaluating the effect of algal biomass incorporation (in various mass ratios) in the SBS polymer matrices, on the physicochemical properties of the final biocomposites. Four mass percentages of the biomass additions were prepared with different masses in different percentages in the mass of the composite. Given that the algal biomass smokes in the extruder at temperatures higher than 180 °C, the working recipes were designed to obtain composites that can be processed at temperatures lower than 180 °C. On the other hand, it was necessary to achieve optimum viscosity of the base material to incorporate algal biomass up to 30%. The recipe for the base polymer styrene-butadiene-styrene composite (SBSC) was set to lead to a Shore A hardness index of around 55 Shore A and a melt flow index high enough to allow the subsequent incorporation of the algal biomass at temperatures below 180 °C. The fluidization of the base composite was achieved by adding paraffin oil.

For the present study, a base polymer composite containing 25% (*w*/*w*) SBS1 copolymer, 50% (*w*/*w*) SBS2 copolymer, and 25% (*w*/*w*) paraffin oil was prepared. The mass ratio SBS1/SBS2 of 1:2 was used so that the paraffin oil could be absorbed by up to 25% (*w*/*w*), as this percentage was found to be optimal for the further incorporation of biomass and for obtaining MSBS samples. The recipe was established after some preliminary tests in order to incorporate as much algal biomass as possible. Reducing the percentage of paraffin oil led to the clogging of extrusion machine when adding higher percentages of algal biomass (up to 30%). It was also found that, for contents exceeding 25%, the paraffin oil did not properly incorporate into the polymer matrix. The final base polymer composite (the blank in all the tests) contained 25% (*w*/*w*) SBS1 copolymer, 50% (*w*/*w*) SBS2 copolymer, and 25% (*w*/*w*) paraffin oil. Preliminary tests performed showed that a percentage of paraffin oil lower than 25% did not allow the incorporation of algal biomass into the intended proportion (up to 30% *w*/*w*). It was also found that for contents exceeding 25%, the paraffin oil did not properly incorporate into the polymer mass. Additionally, the addition of hardeners to the base polymer matrix was avoided because in this case, higher temperatures would have been required for MSBS processing and because, as mentioned before, the higher temperature would cause the algal biomass to smoke or ignite. To obtain microalgae polymer composites, *Arthrospira platensis* and *Chlorella vulgaris* biomass (6.0 ± 0.5% moisture content) were incorporated into the base polymer matrix in various proportions of 5, 10, 20, and 30% (*w*/*w*).

Materials prepared as per the recipes were first homogenized using a heated blade homogenizer (Nanjing Haisi Extrusion Equipment Co. Ltd., Nanjing, China) for uniform mixing and complete dispersion throughout the mixture of solid and liquid powders. First, the raw material is loaded, and next, it passes through a Nanjing Haisi TSE 65 L/D 44 extruder provided with two corotating screws. In the end, the pelletizing operation is performed with an underwater cutting system (water ring).

### 2.3. Polymer Testing Setup

As part of the present study, the obtained polymer composites and control samples were evaluated for their physicomechanical characteristics, their thermal and radiative (ultraviolet) stress, and their behavior in different solvents. Results obtained in these tests may offer valuable information about the potential industrial applications for the studied BB-SBS composites.

To test the physicomechanical characteristics, the composite polymers and the control samples were prepared according to requirements of respective standards, as will be further detailed in Section 2.4. For the tests regarding the thermal, radiative (ultraviolet) stress tests and swelling tests (resistance to chemical solvents/potential extraction of constituents from BB-SBS composite materials), samples were cut into 5 mm cubic test pieces. Thermal behavior tests were performed by maintaining dried polymer samples at 180 °C, 200 °C, and 250 °C for 2 h, separately. Deionized water, ethanol (90%) (*w*/*w*), methanol, and acetone (90%) were chosen as swelling test solvents thanks to their capacity to extract pigments, and a ratio of 5 g polymer sample to 5 mL of solvent was used. Experiments used different immersion times: 24, 48, and 72 h. Structural changes induced by UV radiation were tested by IR spectrometry for samples that were continuously illuminated with 4 UV-C T8 Philips lamps with a power of 30 W, for 80 h. The support for the lamps and the irradiated samples were placed in a closed box with mirror walls in the interior to ensure the continuous, efficient illumination of the samples.

### 2.4. Characterization Techniques

Microscopic investigations were performed on biomass powders, base polymer matrix, and composites by using the scanning electron microscopy technique, with the system SEM-Quanta FEG 250 (Thermo Fischer Scientific, Waltham, MA, USA) in secondary electron mode using the Everhart-Thornley Detector. The samples were not pretreated, powders were used as is, and the base–polymer matrix and composite materials were thinly sliced (2 mm thickness) and mounted on the microscopic stubs, which were fully covered with carbon band.

The sample conditioning for the physicomechanical tests was performed according to EN ISO 291. Specific gravity (6 mm standard plaque) and melting flow index were measured (standard methods SR ISO 2781:2010, C91:2013, SR ISO 1133:2012) with Melt Flow Index Tester (Haida Plastic Melt Flow Index Testing Machine, Dongguan, China). The purpose of the test was to establish the flowability of composites under high temperatures. Sample hardness (Shore A) was measured with Insize Shore A-Durimeter with fixed support according to standard method SR ISO 7619-1:2011. Elongation at break and tensile strength (standard method SR ISO 37:2012) were measured with UTM Computerized Universal Tensile Tester by KMI (Kamal Metal Industries, Ahmedabad, India).

Complementary techniques and corresponding laboratory instruments have been used in this study and will be detailed in the paragraphs below.

Particle size and distribution were measured for the solid biomass powders with Mastersizer 2000, equipped with Scirocco 2000 (Malvern Instruments, Malvern, UK).

The loss from drying (moisture content by heating at 105 °C and differential weighing, to constant mass) and ash content (minerals remained after gradual calcination at 550 °C, to constant mass) for raw materials and composite samples were determined. The procedure consisted of sampling triplicates of 10 g of material (exact mass weighed as described below) in ceramic crucibles that were previously brought to constant mass. First, samples were maintained at 105 °C overnight; then allowed to cool to room temperature into a desiccator and weighed; and then further heated at 105 °C for 1 h, cooled at room temperature, weighed—for several cycles (as needed), up to reaching a constant mass, specifically differences between two weighing measurements of ≤0.0005 g. The result of this step was the loss of moisture content from drying. Next, dried samples were heated with a ramp of 1 h, maintained at 550 °C for 4 h; then allowed to cool at room temperature in a desiccator and weighed; and then further heated to 550 °C for 1 h, allowed to cool at room temperature, weighed—for several cycles (as needed), up to reaching a constant mass of ≤0.0005 g.

For all the measurements involving weighing and thermal treatment, the following instruments were used: semimicro analytical balance (Sartorius Secura 225D-1CEU) with double scales of weighing and readability: 0–120 g with 1·10^−5^ g and 120–220 g with 1·10^−4^ g, and repeatability of 0.03 mg and 0.07 mg respectively; Venticell laboratory oven with forced air circulation, temperature resolution of 1 °C, and uniformity of ±1% (MMM Group, Germany); laboratory muffle furnace L3/11-B180 (Nabertherm, Lilienthal, Germany) with a temperature range from ambient to 1100 °C, temperature resolution of 1 °C, and uniformity of ±5 °C.

Ultraviolet-visible (UV-VIS) spectrometry was used to investigate the pigment migration from the biogenic material into the composites and also from the new polymer biocomposites into four solvents in the chemical stress testing. Evolution 260 BIO (Thermo Fisher Scientific, Waltham, MA, USA) spectrophotometer was used in all experiments; its wavelength range is 190–1100 nm (accuracy ± 0.8 nm) and drift is <0.0005 A/h at 500 nm. For measurements, a spectral bandwidth of 1 nm and quartz cuvettes were used.

Fourier transform infrared (FTIR) spectroscopy was applied as an analytical tool for the characterization of the prepared polymer biocomposites, identification of potential changes when compared to the base polymer matrix, and evaluation of potential structural changes induced by several stress factors (temperature, chemical solvents, oxidative environment, and ultraviolet irradiation). Vertex 80 infrared spectrometer (Bruker, Karlsruhe, Germany) with attenuated total reflection (ATR) system was used for the ease of taking measurements; solid samples without additional preparation could be measured. Spectra were recorded in the mid-infrared wavenumbers region of 4000 cm^−1^ to 400 cm^−1^, with a spectral resolution of 4 cm^−1^; for each result, 32 scans per sample were recorded; and the average spectrum was used for data interpretation. Raw materials and final products were scanned for each studied recipe, and then FTIR spectra of the polymer biocomposites and base polymer matrix were compared to assess potential changes in the molecular structure that may appear owing to the incorporation of the biogenic materials.

The swelling tests were a measure of the net diffusion of polymer, paraffin oil, and microalgae constituents in selected solvents, as well as diffusion of solvent molecules into the polymer matrix.
$$\mathrm{Swelling}\;\mathrm{capacity}\;\left(\%\right)=\frac{m_{t}-m_{0}}{m_{0}}\cdot 100,$$
where *m_t_* is the weight of the sample measured after *t* hours and *m*_0_ is the initial weight of the sample.

## 3. Results and Discussion

The algal raw materials’ chemical compositions are shown in Table 1. The literature data presented in Table 1 are consistent with those indicated by the producer on the batch label.

The experiments showed some differences in the incorporation process of the two types of algal biomass (Chlorella and Spirulina) into the studied base polymer matrix. Thus, it was found that the MSBSC with 30% (*w*/*w*) Chlorella could not be prepared by using the same preparation method and working parameters, as will be further described and as is shown in Figure 1. The experiments showed that during the operations of homogenization and melting at 160 °C, the biomass ignited. However, microalgae composites with 30% (*w*/*w*) Spirulina were successfully prepared. This different heating behavior may be due to the higher content of fats, sugars, and carbohydrates in Chlorella than in Spirulina, as can be observed in Table 1.

### 3.1. Microscopic Characterization and Particle-Size Distribution

Algal biomass particles were characterized with a laser diffraction particle-size analyzer, as presented in Figure 2.

The results of the particle-size analysis showed no significant difference between the two commercially available powders. The particle size ranged from 7 to 120 µm, where most particles were between 9 and 26 µm, for 86% of the total Chlorella particles and 79% of the total Spirulina particles.

The Chlorella and Spirulina biomass looked like clumps of spirals (Figure 3b,c). The SEM image of the SBS composite shows uniform material without structural defects (Figure 3a).

After incorporating the biomass particles into the polymer mass, it was observed that part of the algal biomass particles is fixed in alveoli, where the particles are blocked (Figure 4a–c and Figure 5a–d). The SEM images show the presence of micrometric defects whose number and size increase with the increase in the percentage of incorporated biomass.

### 3.2. Physicomechanical Characterization of SBS Composite

Table 2 presents the physicomechanical data obtained for the blank SBS composites compared with the values for the SBS1 and SBS2 copolymers.

Figure 6 presents the physicomechanical properties of the polymer composite materials. The experiments showed that specific gravity (Figure 6a) increases with the increase in the biomass content in the polymer matrix. Additionally, with regard to specific gravity, small differences were found between the polymers containing Chlorella compared with those composites containing Spirulina. The hardness of the polymer composites (Figure 6c) is close to that of the base composite (control sample). However, it was found that the addition of Spirulina led to an increase in Shore A hardness compared with the polymer mixtures with Chlorella as the biogenic component.

The melt flow index (Figure 6b) is a measure of the ease of flow of the melt of a thermoplastic polymer. The values of melt flow rate are an indirect measure of the ability of the material’s melt to flow under pressure, and it is inversely proportional to the viscosity of the melt in the conditions of the test. For a given material, a higher melt flow index is considered a better characteristic because the molten polymer can be easily formed into the article intended; however, the melt flow index should be low enough that the mechanical strength of the final article will be sufficient for its particular intended use. The experimental findings of this study show that the addition of microalgal biomass causes a decrease in the melt flow rate. The value of the melt flow index for MSBSC (Figure 6b) is around 1.5 g/min. This indicates an increase in the viscosity of the polymer composites thanks to the addition of the algal biomass. The variation in the melt flow index is related to the change in viscosity and the polymer flow rate of different layers in the melted composite. The addition of the algal biomass with particles of different sizes leads to nonhomogeneous material that has a different flow pattern compared with the blank sample elastomer. Thus, particles with a density different from that of the SBSC elastomer can produce turbulent flow, which leads to a reduction in the melt flow index. The melt flow index was lower when the incorporated microalgae biomass was *Arthrospira platensis* compared with polymers that contained *Chlorella vulgaris* powder.

On the other hand, the values of tensile strength and elongation at break decrease when algal biomass is added to the polymer matrix. This characteristic indicates a downgrade of polymer material characteristics compared with the copolymer control samples.

Tensile strength decreases with the addition of algal biomass, increasing with the biomass composition of the composite (Figure 6d). The elongation at break (Figure 6e) for MSBS composites is between 700% and 800%, where the values are lower than those of the SBS composite but quite close to those of SBS copolymer. At the same time, the abrasion resistance (Figure 6f) increases with the increase in biomass content, and the increase seems more significant in the case of the composite that incorporates *Chlorella vulgaris*.

### 3.3. Physicochemical Characterization of SBS Composite

From Figure 7, one may observe that the ash content increases with the increase in the biomass content incorporated into the copolymer matrix, and, as expected, the values are higher than the ash values determined for the control sample. Additionally, these results show a good correlation with the ash content of the algal biomass. Thus, for the powder raw materials of Chlorella, the measured ash content showed values of 4.54 ± 0.74% (*w*/*w*), and for the powder biomass of Spirulina, the measured ash content showed values of 5.71 ± 0.46% (*w*/*w*). The values measured in this study for the ash content of Chlorella and Spirulina biomass are consistent with the values reported in other studies that were focused on the characterization of microalgae biomass [38,39].

During the preparation process of the MSBS composite samples, the polymer granules are melted together with the plasticizer and the microalgae biomass. One of the important parameters needed to assess the effectiveness of the extrusion process is the water content of the resulting composite granules because even a small amount of water may harm the characteristics and performance of the final product, and the formation of surface defects is among the most often encountered. Therefore, determining the water content of the polymer granules before molding is essential. Mass loss on drying is the method frequently used to assess the quality of polymer composites obtained after extrusion.

The values of the mass loss determined for the studied biocomposite polymers were around 3% (*w*/*w*) for all the samples, regardless of the biomass content in the polymer structure (Table 3). This indicates that for the incorporation of the microalgae biomass with a moisture content of around 6% for Chlorella and Spirulina biomass into the polymeric matrices based on SBS copolymers with paraffin oil (up to a mass ratio of 30%), the moisture content of the obtained composites is around 3% for all samples, no matter the percentage of biomass in the polymer matrix.

In Figure 8, the IR spectrum is presented for the SBS composites that have different levels of algal biomass content. It is noticeable that the spectra are similar to the spectrum of the SBS composite. Notable differences appear in the 3100–3600 cm^−1^ area (inset Figure 8), which are related to the O-H groups present in the carbohydrate macromolecules from the algal biomass. The composites with 20% Chlorella presented the highest value of absorbance in the O-H band.

The composites subjected to thermal stress by heating for 2 h at 180 °C, 200 °C, and 250 °C, separately, showed weight loss changes as presented in Table 4. The SBS copolymer that had no biomass incorporated showed a weight loss of up to 2.46% when the respective samples were heated to 250 °C. It was found that at high temperatures, the higher the biomass content, the higher the weight loss values of the polymer composites. This variation might be correlated with two possible processes: first, the volatilization of some compounds that have a boiling point lower than, or equal to, the heating temperatures of the polymer and, second, the oxidation of some chemical species from the algal biomass.

Infrared spectroscopy was used to follow potential variations in the chemical structure of the polymer matrix in relation to the oxidation of the material. Thus, by comparing the infrared (IR) spectra of the polymer composites containing Chlorella biomass incorporated at different ratios with the spectrum of the SBS copolymer as a control sample, no changes were detected, regardless of the algal concentration. On the other hand, the composite polymers containing Spirulina show slight differences in the 1740 cm^−1^ wavenumbers area, which is a specific absorption band for carbonyl groups (Figure 9). This may correlate with either the oxidation of some functional groups in the matrix of the composites and/or the migration of some lipids and/or pigments from the algal granules to the surface of the material.

The studied polymer composites show similar IR spectra when comparing the initial and the final (after the thermal or irradiation stress) samples. Small differences appear in the area of wavenumbers that are characteristic of the O-H stretching band, for those samples that were irradiated for 80 h. Additionally, small differences are observed in the 1720–1740 cm^−1^ area in both testing situations, thermal and radiative stress. There are no such differences in the composite polymer samples without algal biomass incorporated, and they become noticeable starting with the MSBS composites with higher biomass content. Figure 9 shows the IR spectra for the composites with 30% Spirulina before and after the application of thermal and radiative stress; these composites show the most visible spectral changes.

For the studied polymer materials and control/blank sample, Figure 10a–h, presents the swelling capacity variation over the immersion time in four solvents: water, acetone, ethanol, and methanol. The solvents chosen for the extraction proved to be good extractants for the chlorophyll pigments, the color change of the extractant being a good indicator of the mass exchange toward and from the polymer matrix. It is observed that, except for methanol, positive values for the swelling capacity were recorded. In the case of methanol as the contact solvent with the polymer, negative swelling capacity values were found for the blank SBS composite and also for the composite containing 5% Spirulina. The swelling tests were performed by measuring the weights of the polymers between the initial moment and a chosen moment during the experiment. The mass variation of the polymer includes the sum of the two phenomena (loss of mass and solvent uptake). The extractable components from the blank polymer mass (SBSC) are SBS polymer and paraffin oil. On the other hand, the addition of biomass in the elastomer adds to the extractable substances, such as the lipids, proteins, carbohydrates, and pigments contained in the algal biomass. SBS rubber is insoluble in the tested solvents [40], while paraffin oil is insoluble in water and ethanol and is slightly soluble in ethanol and acetone [41]. For the algal components, the most effective extractants for the lipids and pigments are methanol and acetone [42]. In addition to the extraction capacity of these components from the polymer matrix, the swelling capacity of SBR rubber must also be taken into account. SBR can swell the most in the presence of acetone compared water, ethanol, and methanol, which is why the highest solvent uptake values were recorded for acetone [43]. When methanol was used as a solvent for the swelling experiments, the experimental data indicate that the total chemical species that diffuse into the solution have a higher mass than the species that enter the polymer matrix. For all the control samples (SBSC polymer), the weight of the sample decreased when immersed in methanol. This behavior can be attributed to the solubility behavior of the paraffinic oil compared with the three solvents (water, ethanol (90%), and methanol). The highest degree of swelling was recorded in acetone (Figure 10c,d), while the other three solvents, namely water, ethanol, and methanol, exhibit similar behavior. The lowest degree of swelling was found in water. If the degree of swelling for the composites that contain Chlorella is compared with those that contain Spirulina with a similar percentage for the load of the biomass, it was observed that the composites containing Chlorella have higher swelling capacity. The polymers that incorporated Chlorella behave differently than those that contain Spirulina thanks to the different morphology and chemical composition of the biomass. The addition of biomass above 10% leads to materials with a polymer matrix porous structure with a high number of micro alveoli in which the biomass granules were trapped, leading to lower resistance to swelling.

Figure 11a–d and Figure 12a–d show the ultraviolet-visible (UV-Vis) spectra of the resultant solutions from the composite polymer samples swelled in water, acetone, ethanol, and methanol. The UV-Vis spectra of the aqueous solutions show changes in the 350–500 nm domain for all the samples, where the recorded absorbance is different for the samples with different levels of biomass content. The absorbance of the extracted aqueous solution was influenced by the biomass content of the polymeric composite. A higher biomass content leads to the higher absorbance of the extracting aqueous solution. The variation in the absorbance recorded in the UV-Vis spectra had the same tendency regardless of whether the extracting solvent was acetone, ethanol, or methanol. On the other hand, it is observed that the spectra of the extracted solutions with acetone, ethanol, and methanol are similar to the UV-Vis spectra profiles of the chlorophyll pigments and their derivatives [44,45]. Additionally, the UV-VIS spectra recorded for the resulting water solvent after the extraction time is different from the spectra of these pigments. This finding may lead to the conclusion that water and the other studied organic solvents extract different chemical species.

The highest absorbance was recorded for extracting solutions with acetone (Figure 11b and Figure 12b). Additionally, it is observed that in all the situations, shifts in the absorption maxima indicate different compositions of the extract, depending on the biomass concentration added to the polymeric composite. Comparing the absorbance values in the 600–750 nm area (specific to chlorophyll pigments [38,44]) shows that the acetone extracting solution has similar absorbance for the polymer composites containing both types of the studied microalgae biomass. At the same time, the UV-Vis spectra of the ethanol and methanol extracting solutions had higher maximum values recorded for the polymer composite samples with Chlorella.

The study of the swelling capacity variation and the UV-Vis spectra for the extracting solutions correlates the concentration of the extracted pigments with the algal biomass content and the microscopic structure of the polymeric material. It is observed that acetone presents the highest values of swelling capacity, this being the reason for the high maximum absorbance recorded in the UV-Vis spectra (Figure 10b and Figure 11b). Although ethanol and methanol are better extracting solvents for algal biomass pigments [38], the ethanol and methanol extraction solutions obtained from the composite swelling test had lower concentrations of chlorophyll pigments. This experimental finding may be correlated with these solvents’ having a lower capacity to swell the polymer when compared with acetone. As for the water, although it produces a certain swelling in the mass of the polymer, it does not extract pigments, according to the UV-Vis spectra profile.

## 4. Conclusions

This study reported on polymer composites that were prepared by adding *Chlorella vulgaris* biomass and *Arthrospira platensis* biomass, in different ratios, to a base polymer matrix, through an extrusion process. Chlorella biomass and Spirulina biomass are used as food supplements because some consider them “superfoods” because they are high in protein, dietary fibers, vitamins, and complex carbohydrates. Therefore, the algal content of polymer composites may lead to a better composting process for food packaging thanks to a self-disintegrating process where the organic carbon from the polymer chain may act as a feeding source for the microorganisms responsible for plastic disintegration.

The base polymer consisted of an elastomeric composite material made of styrene-butadiene-styrene (SBS) copolymer and paraffin oil with a homogeneous structure and was used as a reference-control sample to compare the newly prepared composites that incorporated algal biomass. The algal biomass was added to the SBS composite in the form of a powder, with particles of different shapes, while their particle sizes were in the micrometer range. It was found that, following the incorporation of the biomass, the resulting polymer composites had acquired an alveolar structure in which the microalgae particles were trapped. Thus, the shape and size of the formed alveoli were similar to the microalgae particles that were added in the extrusion process.

As may be observed in Table 5, the newly obtained materials presented physical and mechanical characteristics close to those of the control sample, namely the SBS composite, and also good resistance to thermal and radiation stress. The swelling studies correlated the extracted pigments with the degree of swelling. Thus, acetone was the best extracting solvent because it had the highest swelling capacity, although according to the studies on the extraction of pigments from the algal biomass, methanol and ethanol were better extracting solvents for chlorophyll pigments. Therefore, the biomass content in the polymer matrix has more of an influence on the behavior of the composites than the base polymer matrix SBS copolymer does, leading to lower resistance to swelling, thermal and radiation stress. In addition, it was found that the polymer composites that incorporated Chlorella behaved differently than those that contained Spirulina because of the distinct morphology and different proportions of the biomass chemical compositions.

## Figures and Tables

**Figure 1 polymers-15-01357-f001:**
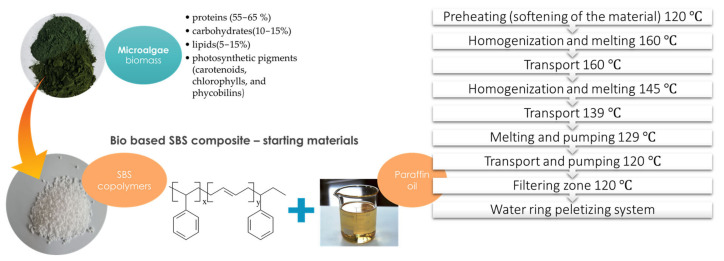
Preparation procedure of MSBS composite preparation.

**Figure 2 polymers-15-01357-f002:**
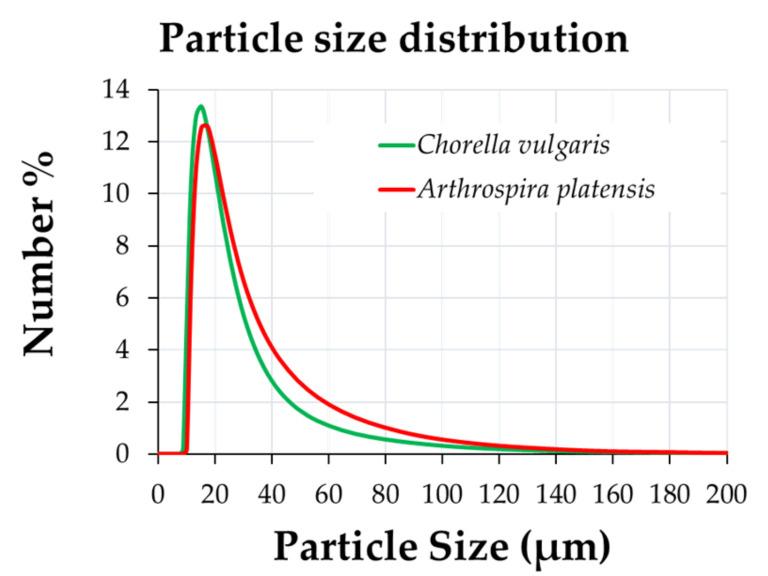
Particle-size distribution of biomass powders used (green: *Chlorella vulgaris*; red: *Arthrospira platensis*).

**Figure 3 polymers-15-01357-f003:**
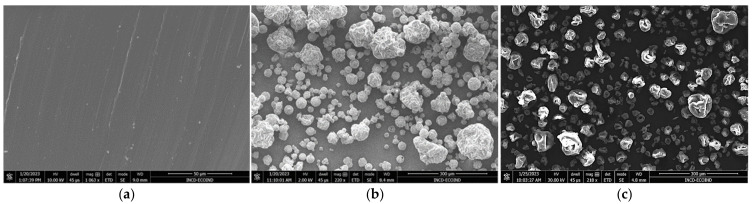
SEM images of (**a**) base polymer matrix; (**b**) *Chlorella vulgaris* biomass powder, and (**c**) *Spirulina platensis* biomass powder.

**Figure 4 polymers-15-01357-f004:**
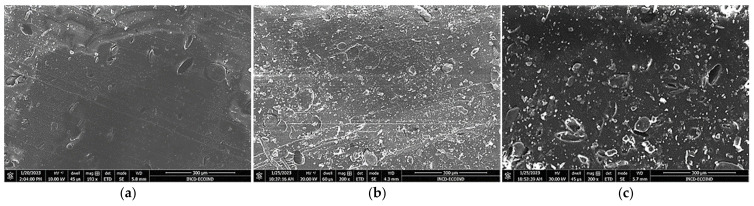
SEM images of polymer MSBS composites with 5% (**a**), 10% (**b**), and 20% (**c**) *Chlorella vulgaris* biomass.

**Figure 5 polymers-15-01357-f005:**
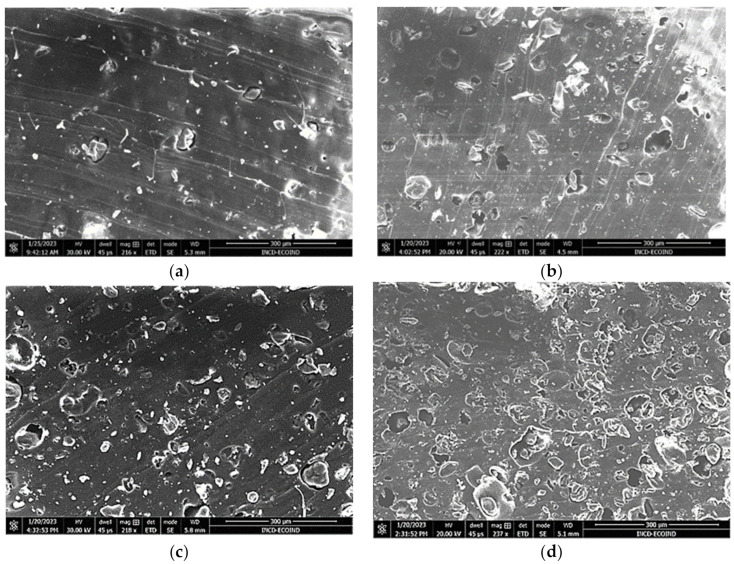
SEM images of polymer MSBS composites with 5% (**a**), 10% (**b**), 20% (**c**), and 30% (**d**) *Arthrospira platensis* biomass.

**Figure 6 polymers-15-01357-f006:**
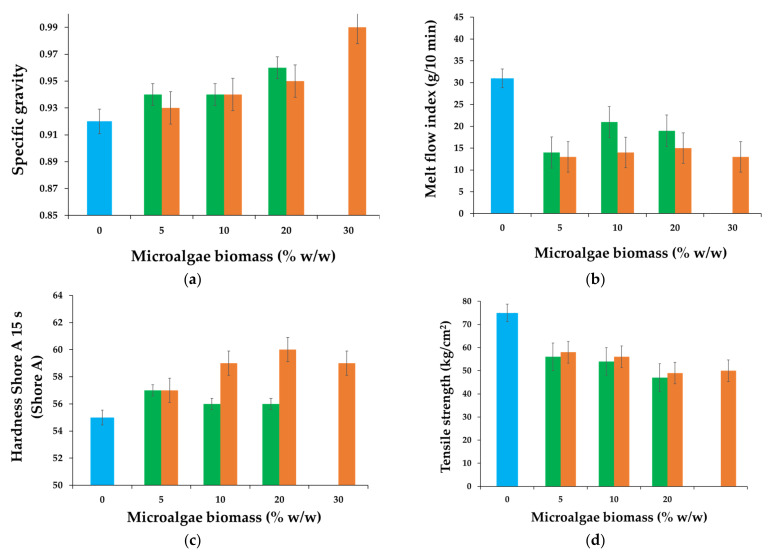
Physicomechanical test results for polymer composites with Chlorella biomass (green bar) and with Spirulina biomass (orange bar), and the control sample–copolymer matrix (blue bar): (**a**) specific gravity, (**b**) melt flow index, (**c**) Hardness Shore A, (**d**) tensile strength, (**e**) elongation at break, and (**f**) abrasion resistance.

**Figure 7 polymers-15-01357-f007:**
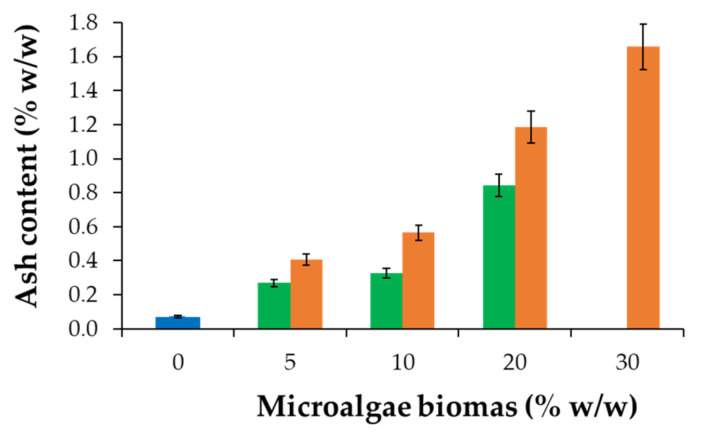
Ash content of MSBS with different content of microalgae: Chlorella (green bar), and Spirulina (orange bar), and in control sample–copolymer matrix (blue bar).

**Figure 8 polymers-15-01357-f008:**
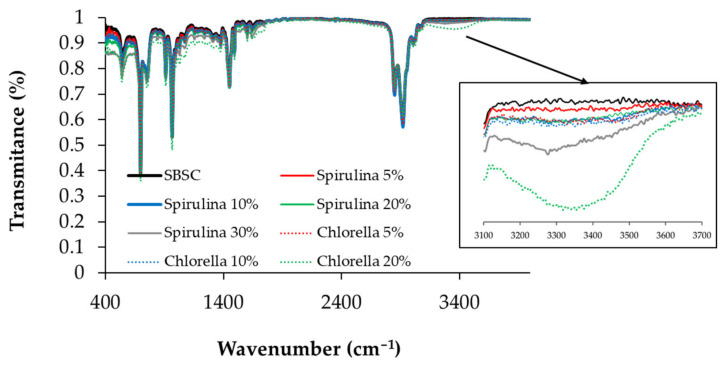
Infrared absorption spectra for SBS composites with different levels of algal biomass content (% *w*/*w*).

**Figure 9 polymers-15-01357-f009:**
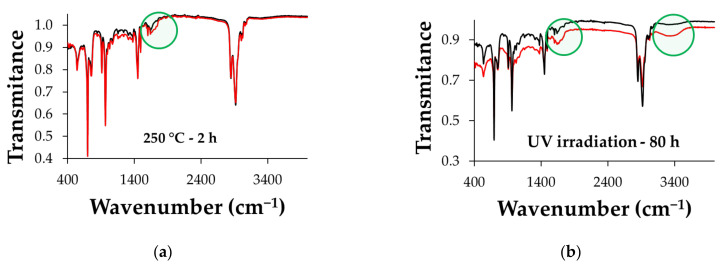
Infrared spectra for SBS composites with 30% Spirulina (% *w*/*w*): (**a**) heated (red line) and unheated (black line) samples, (**b**) irradiated (red line) and nonirradiated (black line) samples.

**Figure 10 polymers-15-01357-f010:**
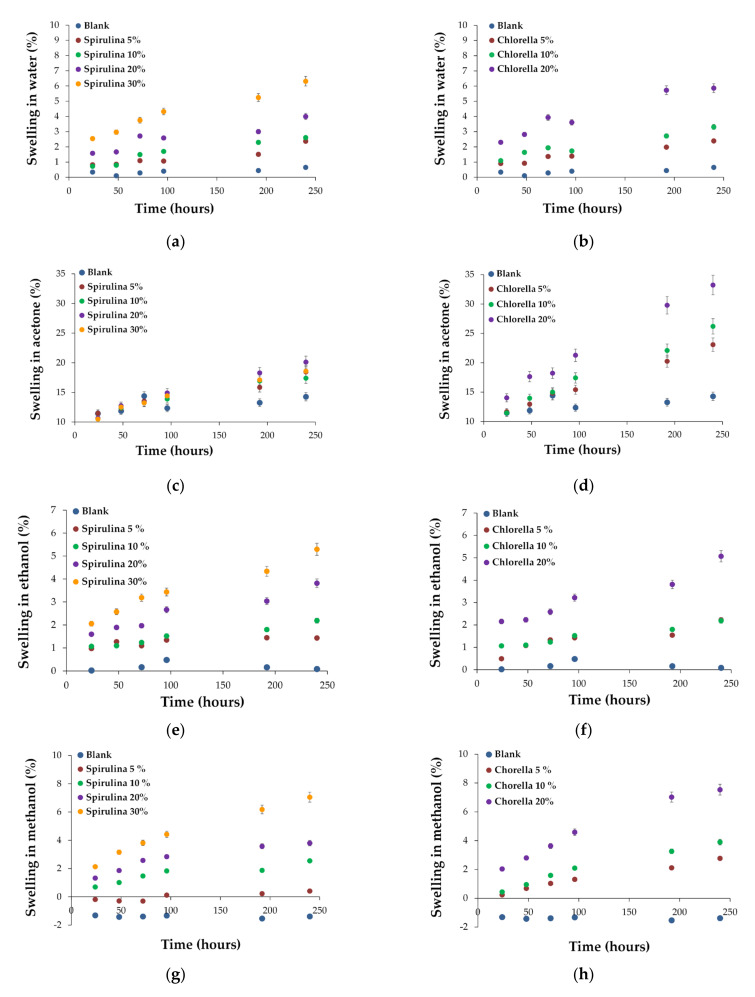
Swelling of MSBS with Chlorella and Spirulina in water (**a**,**b**), acetone (**c**,**d**), ethanol (**e**,**f**), and methanol (**g**,**h**).

**Figure 11 polymers-15-01357-f011:**
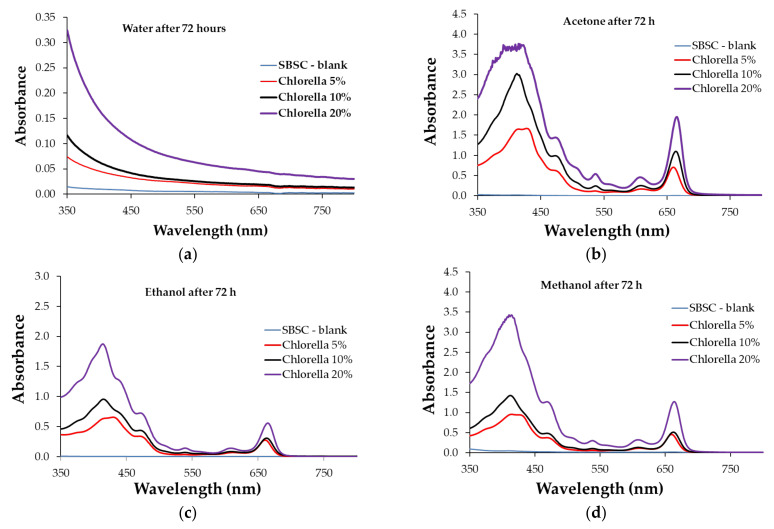
UV-Vis spectra for extraction solutions (swelling tests) after 72 h of contact with Chlorella polymer in (**a**) water, (**b**) acetone, (**c**) ethanol, and (**d**) methanol.

**Figure 12 polymers-15-01357-f012:**
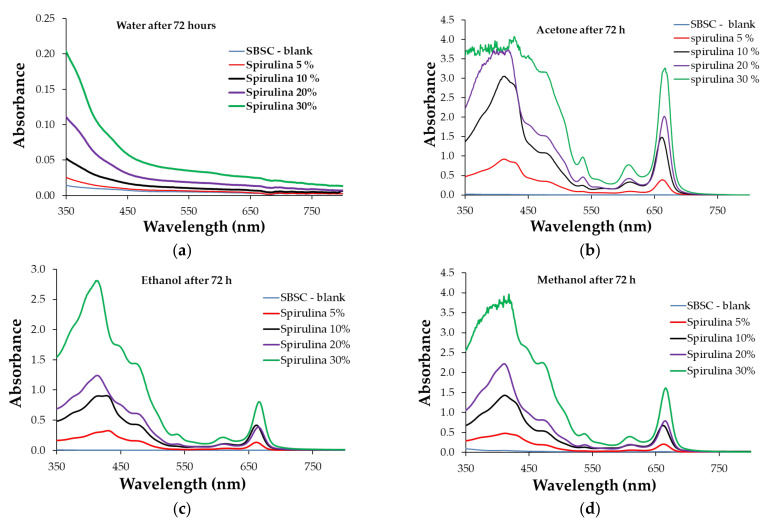
UV-Vis spectra for extraction solutions after 72 h of contact with Spirulina polymer in (**a**) water, (**b**) acetone, (**c**) ethanol, and (**d**) methanol.

**Table 1 polymers-15-01357-t001:** The main macroconstituent content from tested microalgae biomass [33,34,35,36].

	Macroconstituent Content (g/100 g Dry Biomass)		
	Fats	Carbohydrates	Proteins
*Chlorella vulgaris*	10–15	15–25	55–65
*Arthrospira platensis*	5–10	10–20	60–70

**Table 2 polymers-15-01357-t002:** Physicomechanical characteristics of SBS 1 and SBS 2 copolymers vs. SBS composite.

	Copolymers ^1^		SBSC ^2^ SBS1/SBS2/Paraffin Oil (25/50/25, *w*/*w*/*w*)
	SBS1	SBS2	
Styrene-Butadiene Ratio	30/70	40/60	
Tensile strength (kg/cm^2^)	152	203	75
Elongation at break (%)	750	700	1004
Shore hardness (A)	75 ± 7	90 ± 5	55 ± 2
Melt flow rate (g.min^−1^)	0.1–5	0.5–5	2.6–3.6

^1^ literature data [37]; ^2^ measured data.

**Table 3 polymers-15-01357-t003:** Loss on drying (%, *w*/*w*) for MSBS samples with different levels of biomass content.

	Mass Loss on Drying (%, *w*/*w*)				
	0%	5%	10%	20%	30%
*Chlorella vulgaris* MSBS	3.68 ± 0.52	3.91 ± 0.58	3.73 ± 0.31	3.46 ± 0.42	-
*Arthrospira platensis* MSBS		2.88 ± 0.34	2.50 ± 0.16	3.31 ± 0.28	2.96 ± 0.14

**Table 4 polymers-15-01357-t004:** Weight loss (%, *w*/*w*) for MSBS dried samples with different levels of biomass content heated for 2 h at 180 °C, 200 °C, and 250 °C.

		Weight Loss (%, *w*/*w*)				
	Temp. (°C)	0%	5%	10%	20%	30%
*Chlorella vulgaris* MSBS	180 °C	0.61 ± 0.08	1.10 ± 0.09	1.98 ± 0.13	1.78 ± 0.21	-
	200 °C	2.13 ± 0.17	1.25 ± 0.11	3.54 ± 0.21	4.51 ± 0.34	
	250 °C	2.46 ± 0.19	5.79 ± 0.37	4.79 ± 0.29	7.64 ± 0.65	
*Arthrospira platensis* MSBS	180 °C	0.61 ± 0.07	0.92 ± 0.11	1.54 ± 0.11	2.39 ± 0.17	2.86 ± 0.10
	200 °C	2.13 ± 0.13	1.59 ± 0.09	2.52 ± 0.16	3.12 ± 0.21	3.72 ± 0.27
	250 °C	2.46 ± 0.13	5.39 ± 0.42	7.90 ± 0.38	9.89 ± 0.63	9.54 ± 0.67

**Table 5 polymers-15-01357-t005:** Qualifications given to the SBS and MSBS composites, based on experimental recorded data.

Polymer Composite	Specific Gravity	Melt Flow Rate (g·min^−1^)	Tensile Strength (kg/cm^2^)	Elongation at Break (%)	Abrasion Resistance	Solvent Resistance			
						Water	Acetone 90%	Ethanol 90%	Methanol
SBSC									
Chlorella 5%									
Chlorella 10%									
Chlorella 20%									
Spirulina 5%									
Spirulina 10%									
Spirulina 20%									
Spirulina 30%									


	1	2	3	4	5	6	7	8	
	poor							excellent	

## Data Availability

The data presented in this study are available within the present article.

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
