# Peer review of "Preparation and Characterization of Microalgae Styrene-Butadiene Composites Using Chlorella vulgaris and Arthrospira platensis Biomass"

_polymers, 2023, doi:10.3390/polym15061357_

Round 1

Reviewer 1 Report

In the present work, the formulation of composites, based on PBS as polymer matrix and biomass derived from algae as reinforcement, has been described.

Several details must be taken into account prior considering it for publication.

-       The authors describe PBS polymer matrix as a biobased matrix. Please explain why this matrix can be considered as “bio”.

-       The biomass derived from algae has been included in power. The authors describe in line 157 that the biomass is not anhydrous (has moisture content). These data must be moved to 2.1 General chapter due to is a description of the reactants’ composition instead of a methodology.

-       Table 1 refers the values of composition of biomass, including the error of the measurement. Please inform about the nature of the error value: Absolut error, standard deviation, etc…

-       The paragraph from line 163 to 172 describes a result and must be moved to Results chapter.

-       In line 118, the authors inform about the use of “Analytical purity ethanol (96%, evaporation residue ≤ 25 mg/L), acetone (purity ≥ 99.8%, evaporation residue ≤ 3 mg/L), methanol (anhydrous, purity ≥ 99.8%). Later, in line 194, they describe the use of “ethanol 90% (w/w), methanol and acetone (90%)”. Please, if they use a  solution ethanol-water or methanol-water or acetone-water solution are used, please describe.

-       - From line 247 to 249, a procedure of SEM observation has been described. Please inform if the samples were coated (i.e. gold) to induce conductivity.

-       - The order of exposition of materials and methods description is different compared with the order of result emission. This order must be the same to improve understanding.

-       Figure 5/figure 6. The data should be plotted in columns, instead of dots. It will help comparation in a better way.

-       Line 297. The authors report a decrease on melt flow index of the composite due to the incorporation of microalgal biomass without any explanation of the reason. Please discuss this result.

-       Paragraph from 303 to 311. The authors relate a drop in mechanical properties of the composites due to the incorporation of the biomass but they forget to consider the incorporation of the paraffins. Can the authors explain which is the effect of paraffins on the mechanical properties due to the high percentage of incorporation? (higher than 20%, higher than the common values of plasticizers added to polymers). Due to it is an oil, this component can also produce a decrease in the mechanical performance. Please clarify.

-       Paragraph from line 382 to line 384 describes the calculation of solvent uptake. This is a description of a methodology and must be moved to materials and methods.

-       Figure 9 shows the behavior of the composites in the presence of different solvents, having a common trend when using water, ethanol and acetone, but different when using methanol. Using equation 385 the authors describe a swelling process, but observing the data from methanol it is easy to observe that this calculus is incorrect, due to they have a loss of material at the same time of the gain of mass due to the solvent. So, the authors should separate both effect, loss of material (on one side) and solvent uptake (on another). By this way they could differentiate the different behavior of the solvents. I suggest to plot loss of mass due to solubilization of the material in the medium, and in another graph the gain of mass due to the solvent uptake.

Author Response

Dear Reviewer, thank you very much for your time and your thorough analysis.

In the following paragraphs the response to your observations is presented point by point.

-       The authors describe PBS polymer matrix as a biobased matrix. Please explain why this matrix can be considered as “bio”.

Thank you for this remark. Obviously, we do not consider the polymer matrix as “bio”. However, the manner we presented the information in the rows 49 – 75 (in the uploaded manuscript version) induced this incorrect idea. The text between the quoted rows was separated in distinct paragraphs, so that a better (and correct) understanding to the readers is allowed in this form. In addition, a supplementary phrase was added to avoid confusions: “Both of them are fossil and non-biodegradable polymers.”

-       The biomass derived from algae has been included in powder. The authors describe in line 157 that the biomass is not anhydrous (has moisture content). These data must be moved to 2.1 General chapter due to is a description of the reactants’ composition instead of a methodology.

The suggestion was fulfilled by moving the information to the section 2.1.. 

-       Table 1 refers the values of composition of biomass, including the error of the measurement. Please inform about the nature of the error value: Absolut error, standard deviation, etc…

A supplementary information was added in the caption of Table 1 to clarify the nature of indicated error: “[values ± standard deviations]”.

-       The paragraph from line 163 to 172 describes a result and must be moved to Results chapter.

Paragraphs from row 159 to 172 was moved to section Results, according to reviewer suggestion.

-       In line 118, the authors inform about the use of “Analytical purity ethanol (96%, evaporation residue ≤ 25 mg/L), acetone (purity ≥ 99.8%, evaporation residue ≤ 3 mg/L), methanol (anhydrous, purity ≥ 99.8%). Later, in line 194, they describe the use of “ethanol 90% (w/w), methanol and acetone (90%)”. Please, if they use a  solution ethanol-water or methanol-water or acetone-water solution are used, please describe.

Thank you for this remark, indeed, we prepared 90% solutions of the three organic solvents. We inserted a clarification phrase as suggested (in section 2.1.): “namely 90% (v/v) solutions of the three organic solvents”.

-       From line 247 to 249, a procedure of SEM observation has been described. Please inform if the samples were coated (i.e. gold) to induce conductivity.

The paragraph describing sem measurements was modified in the manuscript:

 “Microscopic investigations were performed on biomass powders, base-polymer matrix and composites using the scanning electron microscopy technique, with the system SEM-Quanta FEG 250 (Thermo Fischer Scientific, USA) in secondary electrons mode using the Everhart-Thornley Detector. The samples were not pre-treated, powders were used as is while the base polymer and composite materials were thin sliced (2 mm thickness) and mounted on the microscopic stubs fully covered with carbon band.”

-       The order of exposition of materials and methods description is different compared with the order of result emission. This order must be the same to improve understanding.

The experimental section was modified as indicated by the reviewer.

-       Figure 5/figure 6. The data should be plotted in columns, instead of dots. It will help comparation in a better way.

The figures were modified as suggested. Thank you for the observation.

-       Line 297. The authors report a decrease on melt flow index of the composite due to the incorporation of microalgal biomass without any explanation of the reason. Please discuss this result.

An explanation paragraph was added in the manuscript:

“The variation of the melt flow index is related with the change in viscosity and the polymer flow rate of different layers in the melted composite. The addition of algal biomass with particles of different sizes leads to non-homogeneous material that has a different flow pattern compared to the blank sample elastomer. Thus, particles with a density different from that of the SBSC elastomer can produce turbulent flow, which leads to a reduction in the melt flow index.”

-       Paragraph from 303 to 311. The authors relate a drop in mechanical properties of the composites due to the incorporation of the biomass but they forget to consider the incorporation of the paraffins. Can the authors explain which is the effect of paraffins on the mechanical properties due to the high percentage of incorporation? (higher than 20%, higher than the common values of plasticizers added to polymers). Due to it is an oil, this component can also produce a decrease in the mechanical performance. Please clarify.

Thank you for the observation. The information is not clear enough presented in the manuscript.

In the experimental section, the preparation method of the control sample is presented, which is a composite of SBS and paraffinic oil. The mechanical tests were performed using samples of the reference composite material, which is made with paraffin oil. The results for the composite without algal biomass are  specific to the elastomer in which paraffinic oil is added.

The text on how to prepare composites has been modified so that the information is clearer.

 “For the present study a base-polymer composite contained 25% (w / w) SBS1 copolymer, 50% (w / w) SBS2 co-polymer and 25% (w / w) paraffin oil was prepared. The mass ratio SBS1 / SBS2 of 1:2 was used, so that the paraffin oil could be absorbed up to 25% (w / w), as this percentage was found to be optimal for further incorporation of biomass and obtaining BBSBS samples. The recipe was established after some preliminary tests in order to incorporate as much algal biomass as possible. Reducing the percentage of paraffinic oil led to clogging of extrusion machine when adding higher percentages of algal biomass (up to 30%). It was also found that, for contents exceeding 25%, the paraffin oil did not properly incorporate in the polymer matrix. The final base-polymer composite (the blank in all the tests) contained 25% (w / w) SBS1 copolymer, 50% (w / w) SBS2 co-polymer and 25% (w / w) paraffin oil. Preliminary tests performed showed that a percentage of paraffin oil lower than 25% did not allow the incorporation of algal biomass in the intended proportion (up to 30% w/w). It was also found that, for contents exceeding 25%, the paraffin oil did not properly incorporate in the polymer mass.”

-       Paragraph from line 382 to line 384 describes the calculation of solvent uptake. This is a description of a methodology and must be moved to materials and methods.

The paragraph was moved in the indicated section.

-       Figure 9 shows the behavior of the composites in the presence of different solvents, having a common trend when using water, ethanol and acetone, but different when using methanol. Using equation 385 the authors describe a swelling process, but observing the data from methanol it is easy to observe that this calculus is incorrect, due to they have a loss of material at the same time of the gain of mass due to the solvent. So, the authors should separate both effect, loss of material (on one side) and solvent uptake (on another). By this way they could differentiate the different behavior of the solvents. I suggest to plot loss of mass due to solubilization of the material in the medium, and in another graph the gain of mass due to the solvent uptake.

Indeed, the paragraph explaining the swelling experiment in methanol is not clear enough. Regarding this observation the experimental data showed that at all the control samples (SBSC polymer) had a decrease of mass when immersed in methanol. The following paragraph was added in the manuscript to respond this request. 

“The swelling tests were performed by measuring the weights of the polymers between the initial moment and a chosen moment during the experiment. The mass variation of the polymer includes the sum of the two phenomena (loss of mass and solvent uptake). The extractable components from the blank polymer mass (SBSC) are SBS polymer and paraffin oil. On the other hand, the addition of biomass in elastomer adds to the extractable substances lipids, proteins, carbohydrates and pigments contained in algal biomass. SBS rubber is insoluble in the tested solvents [40], while paraffin oil is insoluble in water and ethanol, and slightly soluble in ethanol and acetone [41]. For the algal components, the most effective extractants for lipids and pigments are methanol and acetone  [42]. Besides the extraction capacity of these components from the polymer matrix, the swelling capacity of SBR rubber must also be taken into account. SBR can swell the most in the presence of acetone compared water, ethanol and methanol, this being the reason why the highest solvent uptake values were recorded for acetone [43]. When methanol was used as solvent for swelling experiments, the experimental data indicates that the total chemical species that diffuse into the solution have a higher mass than the species that enter the polymer matrix. At all the control samples (SBSC polymer) the weight of the sample decreased when immersed in methanol. This behavior can be attributed to the solubility behavior of the paraffinic oil compared to the three solvents (water, ethanol 90% and methanol).”

Reviewer 2 Report

I congratulate the authors for the theme developed and the scope of the work presented. However, the submitted manuscript is quite incomplete in some chapters and requires some corrections regarding the presentation of results.

There are some comments for the authors.

Keywords

_Must put "biogenic materials" as one of the keywords.

Introduction

_I ask that authors review lines 40-50. All the text written here is very similar to other works already published. Please review carefully.

Materials and methods

_Line 134-161 - In sub-chapter 2.2, they present a series of conclusive results for preparing the composites; it is not explained or detailed how they obtained them (they are only given!). Please review and complete it.

_Line 214-216 - Describe the methods used. Please complete.

Results and discussion

_Line 264 - replace fig. 2a per fig. 3a.

_In subchapter 3.1, there are two figures with the number 3 (line 260 and line 266). Please check.

_Line 289 - In fig 5, you must put the legend of the colors inscribed in each graph presented.

_Line 334 - In table 3, the formatting is not correct. They should show "Humidity (%, w/w) for" in the different percentages line. Please check.

Conclusion

_Authors must present a table in which they inscribe the different results obtained to make them easier to read.

Author Response

Dear Reviewer, thank you very much for your kind words and your observations

In the following paragraphs the response to your observations is presented point by point.

  1. Keywords

_Must put "biogenic materials" as one of the keywords.

Thank you for this remark, the keywords were added.

  1. Introduction

_I ask that authors review lines 40-50. All the text written here is very similar to other works already published. Please review carefully.

The paragraph was fully rephrased.

  1. Materials and methods

_Line 134-161 - In sub-chapter 2.2, they present a series of conclusive results for preparing the composites; it is not explained or detailed how they obtained them (they are only given!). Please review and complete it.

Thank you for this remark. Though we detailed the full procedure for the composite preparation in lines 173 – 180, we understand that is not clear enough. So, we moved the text from lines 173-180, and included it just after the line 137, so that we now hopefully offer a better understanding to the readers.

_Line 214-216 - Describe the methods used. Please complete.

Details regarding the working procedures were added in the indicated paragraph:

“The procedure involved consisted in sampling triplicates of 10 g of material (exact masses weighed as described below) in ceramic crucibles that were previously brought to constant mass. First, samples were maintained at 105 °C overnight, then allowed to cool to room temperature into a desiccator, weighted, then further heated at 105 °C for 1 hour - cooled at room temperature – weighted, for several cycles (as needed), up to reaching a constant mass, namely differences between two weighing measurements ≤ 0.0005 g. The result of this step was the loss on drying / moisture content. Then, dried samples were heated with a ramp of 1 hour, maintained at 550 °C for 4 hours, allowed to cool at room temperature in a desiccator, weighted, and then further heated to 550 °C for 1 hour - allowed to cool at room temperature – weighted, for several cycles (as needed), up to reaching constant mass ≤ 0.0005 g.”

  1. Results and discussion

_Line 264 - replace fig. 2a per fig. 3a.

Correction has been done as indicated, thank you.

_In subchapter 3.1, there are two figures with the number 3 (line 260 and line 266). Please check.

Thank you for the observation. The figures were renumbered.

_Line 289 - In fig 5, you must put the legend of the colors inscribed in each graph presented.

The graphs from figure 5 were replaced with bar graphs that are more suggestive. Legend of the colors was mentioned in the figure caption as requested.

_Line 334 - In table 3, the formatting is not correct. They should show "Humidity (%, w/w) for" in the different percentages line. Please check.

Thank you for this observation. According to suggestion, Table 3 was reorganized for a better clarity.

  1. Conclusion

_Authors must present a table in which they inscribe the different results obtained to make them easier to read.

A table with qualifications was introduce in the conclusion section as indicated.

Round 2

Reviewer 1 Report

After the first round I still have some minor remarks that must be attended.

About the consideration of the polymer matrix as "bio" I agree with the explanation that the authors have given, but still I can read in the document BBSBS (biobased styren-butadien-styren) composite. In my opinion these composites cannot be considered "bio". Of course reinforcement has a natural origin , but not for the matrix. So, I suggest to remove "BB" from BBSBS. I suggest to include some qualifications such as Biobased reinforcement o something like this, but no biobased composites.

About the paraffin added to the formulations, I consider that the incorporation in a quantity higher that 20% will produce exudations from the material. In line 183, the authors inform about 25% as optimal. I suggest to inform what happens if they add less and what happens when they add more. In another words, which is the criterium to consider "optimal" such composition. 

Author Response

Dear Reviewer,

Thank you for your observations.

We made some corrections to the manuscript as suggested:

  1. About the consideration of the polymer matrix as "bio" I agree with the explanation that the authors have given, but still I can read in the document BBSBS (biobased styren-butadien-styren) composite. In my opinion these composites cannot be considered "bio". Of course reinforcement has a natural origin , but not for the matrix. So, I suggest to remove "BB" from BBSBS. I suggest to include some qualifications such as Biobased reinforcement o something like this, but no biobased composites.
  • The composites were renamed as MSBS (microalgae styrene butadiene styrene) composites. The title was changed and the BBSBS abbreviation was replaced with MSBS in the manuscript.
  1. About the paraffin added to the formulations, I consider that the incorporation in a quantity higher that 20% will produce exudations from the material. In line 183, the authors inform about 25% as optimal. I suggest to inform what happens if they add less and what happens when they add more. In another words, which is the criterium to consider "optimal" such composition. 
  • The paraffine oil was added to SBS to make the compound ready to incorporate the microalgae biomass without exudation. If the paraffin oil was less than 25 % the mixture with microalgae “smoked” and the extrusion machine gets clogged. If the paraffin oil was added more than 25 % the blank sample (SBS composite) had “wet” surface because of the paraffin oil added in the composite.

A paragraph was added in the manuscript to explain the percentage choosed for the mixture:

“For the present study a base-polymer composite contained 25% (w / w) SBS1 copolymer, 50% (w / w) SBS2 co-polymer and 25% (w / w) paraffin oil was prepared. The mass ratio SBS1 / SBS2 of 1:2 was used, so that the paraffin oil could be absorbed up to 25% (w / w), as this percentage was found to be optimal for further incorporation of biomass and obtaining MSBS samples. The recipe was established after some preliminary tests in order to incorporate as much algal biomass as possible. Reducing the percentage of paraffin oil led to clogging of extrusion machine when adding higher percentages of algal biomass (up to 30%). It was also found that, for contents exceeding 25%, the paraffin oil did not properly incorporate in the polymer matrix. The final base-polymer composite (the blank in all the tests) contained 25% (w / w) SBS1 copolymer, 50% (w / w) SBS2 co-polymer and 25% (w / w) paraffin oil. Preliminary tests performed showed that a percentage of paraffin oil lower than 25% did not allow the incorporation of algal biomass in the intended proportion (up to 30% w/w). It was also found that, for contents exceeding 25%, the paraffin oil did not properly incorporate in the polymer mass.”

Reviewer 2 Report

I thank the authors for their care, dedication and effort in responding to comments. The final manuscript has a much higher scientific quality than the initial one. I think that the manuscript has the required quality for publication.

Author Response

Dear Reviewer, 

Thank again for your time and your observations that improved the presentation of the study.